# Lynch Syndrome and Thyroid Nodules: A Single Center Experience

**DOI:** 10.3390/genes15070859

**Published:** 2024-06-30

**Authors:** Irene Spinelli, Simona Moffa, Francesca Fianchi, Teresa Mezza, Francesca Cinti, Gianfranco Di Giuseppe, Clelia Marmo, Gianluca Ianiro, Francesca Romana Ponziani, Annalisa Tortora, Maria Elena Riccioni, Andrea Giaccari, Antonio Gasbarrini

**Affiliations:** 1Centro Malattie Apparato Digerente (CEMAD), Fondazione Policlinico Universitario Agostino Gemelli IRCCS, 00168 Rome, Italy; 2Centro per le Malattie Endocrine e Metaboliche, Fondazione Policlinico Universitario Agostino Gemelli IRCCS, 00168 Rome, Italy; 3Endoscopia Digestiva Chirurgica, Fondazione Policlinico Universitario Agostino Gemelli IRCCS, 00168 Rome, Italy; 4Gastroenterologia B, Azienda Ospedaliera Universitaria Integrata Verona, 37126 Verona, Italy

**Keywords:** lynch syndrome, thyroid nodules, ultrasound

## Abstract

Background: Lynch syndrome (LS) is a genetic disease with increased risk of colorectal cancer and other malignancies. There are few reported cases of thyroid cancer in LS patients. The aim of this study is to investigate the presence of thyroid nodules in LS patients and to explore their association with the genetic features of the disease. Methods: A retrospective and descriptive analysis was conducted to include all LS patients followed at the CEMAD (Centro Malattie Apparato Digerente) of Fondazione Policlinico Universitario A. Gemelli IRCCS. The characteristics of LS disease, gene mutations, and previous history of thyroid disease were evaluated. Majority of patients underwent thyroid ultrasound (US), and nodule cytology was performed when needed. Results: Of a total of 139 patients with LS, 110 patients were included in the study. A total of 103 patients (74%) underwent thyroid ultrasound examinations, and 7 patients (5%) had a previous history of thyroid disease (cancer or multinodular goiter). The mean age was 51.9 years. Thyroid nodules were found in 62 patients (60%) who underwent US, and 9 of them (14%) had suspicious features of malignancy, inducing a fine-needle aspiration biopsy. A cytologic analysis classified 7 of 9 cases (78%) as TIR2 and 2 (22%) as TIR3a. Between patients with nodular thyroid disease (single nodule, multinodular goiter, and cancer), most of them (25 patients, 36% of total) were carriers of the MSH6 mutation, while 22 (32%), 17 (24%), and 5 (7%) had MSH2, MLH1, and PMS2 mutations, respectively. Conclusions: A high prevalence of thyroid nodules was found in patients with LS, especially in MSH6-carrying patients. Performing at least one thyroid ultrasound examination is suggested for the detection of nodular thyroid disease in LS patients. Systematic investigations are needed to estimate their prevalence, features, and risk of malignant transformation.

## 1. Introduction

Lynch syndrome (LS) is an autosomal dominant inherited disease caused by mutations in the mismatch repair (MMR) genes MLH1, MSH2, MSH6, and PMS2, and the EPCAM gene, which regulates MSH2 expression.

LS patients are at an increased risk, at a young age and before the national screening period, of colorectal cancer (CCR) and other extracolonic malignancies, such as endometrial, ovarian, upper gastrointestinal (GI), urinary tract, skin, breast, and prostate cancer.

Cancer development risk is gene-specific and depends on the underlying pathogenic variant in the mismatch repair (MMR) gene. Consequently, clinical management varies based on this gene-specific risk.

Identification of patients with LS is important to start a lifetime surveillance program in order to guarantee early detection of cancer [1,2,3,4]. 

Other types of cancer, which are not common for this syndrome and whose etiologies are still controversial, can be diagnosed in LS patients. It is currently under debate whether these tumors are coincidental or part of the LS tumor spectrum. 

Thyroid cancer (TC) is not considered part of Lynch syndrome, but there are a few reported cases of TC in patients suffering from LS, mostly associated with MLH1 and MSH2 germline mutation [5,6,7,8].

Thyroid cancer occurs in 7–15% of thyroid nodule cases, which are very frequent in the general population, accounting for about 20–50% of prevalence, according to the method of detection and the subject’s age [9,10]. The major task in their management is to detect the minority of cases that correspond to a malignant lesion.

In the scientific literature, there are no data about the prevalence of thyroid nodules in Lynch syndrome. 

The aim of our study is to investigate the presence and features of thyroid nodules in LS patients, assess their cytologic classes, and explore their association with the genetic features of the disease. 

## 2. Materials and Methods

A retrospective and descriptive study was conducted to include all LS patients followed at CEMAD (Centro Malattie Apparato Digerente) of Fondazione Policlinico Universitario A. Gemelli IRCCS between January 2018 and January 2024. Lynch syndrome diagnosis was made by a direct germline test analyzing the genomic regions of colorectal cancer susceptibility gene panel in leukocyte DNA.

Patient selection was performed by querying the database of Rare Gastrointestinal and Liver Diseases of the CEMAD (Centro Malattie Apparato Digerente) Department and extracting all cases who underwent thyroid ultrasound or with previous thyroid disease history. 

Patients who did not undergo thyroid US and without previous thyroid disease history were not included. 

We reviewed the clinical history of patients with previous thyroid disease, which included multinodular goiter and thyroid cancer. All these patients underwent thyroidectomy. Pathology reports of the surgery were studied. 

All patients enrolled were previously referred for a thyroid-screening US performed by endocrinologists. 

In our center, a Color and Power Doppler US scan is usually performed using an ultrasound (Esaote Mylab A50, Esaote SPA, Firenze, Italy) equipped with a high-resolution linear probe (11 MHz). 

Reports and images were reviewed, and all relevant information was extracted and tabulated. 

According to the ultrasonographic classification of the “American Thyroid Association” [9] and the Italian Consensus for DTC management [10], we stratified thyroid nodules as follows: Class 1: Low-risk thyroid lesion: purely cystic nodules, mostly cystic (>80%) nodules with reverberating artifacts, spongiform, or isoechoic or hyperechoic nodules not associated with suspicious US findingsClass 2: Intermediate-risk thyroid lesion: slightly hypo- or isoechoic nodules with ovoid-to-round shape and smooth or ill-defined margins. May be present: intra-nodular vascularization, macro- or continuous-rim calcifications, increased stiffness at elastography, or hyperechoic spots of uncertain significanceClass 3: High-risk thyroid lesion: nodules with at least one of the suspicious findings: marked hypo-echogenicity, spiculated or micro-lobulated margins, micro-calcifications, taller-than-wide shape, or extrathyroidal growth or lymphadenopathy

A US-guided fine-needle aspiration (FNA) was previously performed for all high-risk thyroid nodules with a diameter ≥ 10 mm, and in selected cases of nodules with a diameter of 5–9 mm. For intermediate-risk and low-risk nodules, an FNA was performed for nodules with a diameter ≥ 20 mm.

The cytological diagnosis was classified according to the “Italian consensus for the classification and reporting of thyroid cytology” [10] as follows:TIR 1: Non-diagnosticTIR 2: Non-malignant/benignTIR 3: A Low-risk indeterminate nodulesTIR 3B: High-risk indeterminate lesionsTIR 4: Suspicious for malignancyTIR 5: Malignant

Patients not selected to perform FNA were monitored with US scans at 6 or 12 months as clinically appropriate.

For each included patient, age, pathogenic germline variants, affected branch of the family, comorbidities, and other neoplastic diseases were evaluated. 

Each thyroid phenotype was associated with a specific gene mutation. 

### Statistical Analysis

Categorical variables are expressed as counts and percentages; continuous variables are expressed as means and standard deviations (SDs). 

Data were analyzed using Excel (16.77.1) for Windows and SPSS software (SPSS 29—September 2022).

## 3. Results

A total of 139 LS patients, 52 males and 87 females, were considered for the study.

Of these 139 patients, 110 were eligible for the study. 103 patients (74%) underwent thyroid ultrasound examination, and 7 patients (5%) had a previous history of thyroid disease (cancer or multinodular goitre). A total of 29 patients (21%) had not already undergone US at the time of analysis, so they were excluded from the study.

We only considered the 110 patients who underwent the thyroid US and had a previous history of thyroid disease. Demographic data are reported in Table 1.

For these patients, 35 males and 75 females, the mean age was 48.6 ± 12.3 years (SDs).

We also performed an analysis dividing patients in two subgroups, one including patients with thyroid nodules and previous history of thyroid disease (69 patients) and one including patients who did not present thyroid nodules in the ultrasound (US) (41 patients).

Groups of patients are shown in Figure 1.

### 3.1. Patients Who Underwent Thyroid US

A total of 103 out of 139 patients (74%) in our cohort underwent thyroid ultrasound examination.

A total of 41 patients (40%) had nodules absent in the US.

Thyroid nodules were found in 62 patients (60%) who underwent US, 46 were females and 16 were males. Of those, 22 (35%) had a single nodule and 40 (65%) had multiple nodules.

A total of 53 patients (86%) had nodules without suspicious features of malignancy (Class 1), and 9 patients (14%) had nodules with suspicious features of malignancy. Of the 9 patients, 6 (9.5%) had intermediate-risk and 3 (4.5%) had high-risk thyroid lesions (Class 2 and 3), inducing a fine-needle aspiration biopsy.

Cytologic analysis classified 7 of these 9 cases (78%) as TIR2 and 2 (22%) as TIR3a. Patients with TIR2 thyroid cytology were referred for ultrasound follow-up at 12 months, while both TIR3a patients were re-examined with a new fine-needle aspiration at 6 months.

### 3.2. Patients with Previous Nodular Thyroid Disease

Data emerging from their previous clinical history showed that 7 of 139 patients (5%) had a previous history of thyroid disease (cancer or multinodular goiter). Out of the 7, a total of 2 (28%) had thyroid cancer and both underwent total thyroidectomy, with or without neck nodal dissection depending on the stage of disease. A histopathological analysis showed papillary carcinoma in both cases. A total of 5 patients (72%) underwent total thyroidectomy for multinodular goiter and they are followed annually only for the management of levothyroxine replacement therapy.

### 3.3. Associated Neoplastic Diseases

Of the 110 patients who underwent thyroid US and had a previous history of thyroid disease, associated neoplastic diseases were evaluated. The data are shown in Table 2.

We compared the data between two groups, one of 69 patients who presented thyroid nodules in US and with a previous history of thyroid disease and one patient without thyroid nodules in US (41 patients).

### 3.4. Mutation Evaluation

For the subgroup of 69 patients who presented thyroid nodules in the US and had a previous history of thyroid disease, a review of pathogenic germline variants showed that most of them (25 patients, 36% of total) were carriers of an MSH6 germline mutation, while 22 (32%), 17 (24%), and 5 (7%) had MSH2, MLH1, and PMS2 mutations, respectively.

In particular, of the seven patients with intermediate and high risk nodules, one (15%) patient was carrying the MSH2 mutation, two (28%) were carrying the MLH1 mutation, and the two patients who previously had thyroid cancer were carrying the MSH6 and MSH2 mutations, respectively.

For the subgroup of 41 patients who did not present thyroid nodules, 12 (29%) patients were carriers of the MSH6 mutation, 17 (41%) of MSH2, 11 (27%) of MLH1, and 1 (3%) of PMS2.

## 4. Discussion

Lynch syndrome is an autosomal dominant inherited disease caused by mutations in the mismatch repair genes which cause an increased risk of developing colorectal cancer and other extracolonic malignancies at a relatively young age.

Specific neoplastic risk, clinical management, and lifetime surveillance programs depend on the MMR gene’s underlying pathogenic variant.

Other types of tumors can be diagnosed in LS patients, but with a lower frequency. For example, thyroid cancer is neither usually considered to be part of the Lynch syndrome spectrum nor to have its gene mutations been reported in thyroid malignancies [11,12,13]. On the contrary, TC is strongly associated with several other genetic syndromes such as PTEN hamartoma tumor syndrome (PHTS), Cowden’s disease, familial adenomatous polyposis (FAP), Carney complex, multiple endocrine neoplasia (MEN) 2, Werner syndrome/progeria and others, which warrant a screening based on various components of that syndrome [9,14,15].

Only few cases of TC in Lynch syndrome are reported in the literature [5,6,7,8]. Because of the lack of data regarding thyroid nodules and cancer in Lynch syndrome, we investigated their presence and features, assessing their cytologic classes and the association with the genetic mutation responsible for the disease.

Of our cohort of 139 LS patients, we included only 110 patients who had undergone thyroid US or had a previous history of thyroid disease. Even two had a history of thyroid papillary carcinoma. Of these 110 patients, we identified and compared two subgroups, one including 69 patients with thyroid involvement (nodules and previous history of thyroid disease) and one including patients without thyroid nodules in the ultrasound (41 patients).

The prevalence of thyroid nodules in LS patients was 60%, exceeding that of the general population (20–50%) [10]. Only 14% of these cases had suspicious features of malignancy, inducing a fine-needle aspiration biopsy, of which most of them (78%) had TIR2, and 22% had TIR3a.

We also evaluated associated neoplastic diseases for both subgroups with evidence of major neoplastic involvement in patients with thyroid nodules and with a previous history of thyroid disease (Table 1) compared to patients without thyroid nodules.

In addition, we reviewed pathogenic germline variants for each subgroup. Most of the patients with thyroid nodules in the US and a previous history of thyroid disease were carriers of MSH6 mutation, while in the subgroup without thyroid nodules, the most frequent mutation found was MSH2. The two patients with previous thyroid cancer were carriers of MSH6 and MSH2 mutation, respectively.

Although thyroid cancer is not considered a part of Lynch syndrome, several reports and our study suggest that these two conditions may coexist without revealing their potential association [16,17]. Mutations in BRAF, RET/PTC, NTRK1, or RAS have been reported in up to 70% of differentiated thyroid cancers [18,19] and historically it does not depend on an MMR deficiency [20,21,22,23]. However, some authors found an MLH1 and MSH2 protein impairment in thyroid cancer tissues, a condition that could increase the chance of somatic genetic alteration [5,6,7,8,19].

In addition, due to several technical mechanisms, false negatives of immunohistochemistry analysis may occur, wrongly excluding mismatch repair defects and synchronous cancers linked to the same pathogenic process [24]. These findings suggest that thyroid cancer in Lynch syndrome may be not incidental, but most likely it may develop in association with the underlying germline defect. Unfortunately, genetic analysis of the neoplastic tissue was not performed for either of our thyroid cancer cases.

Alternatively, as mentioned by Fazekas-Lavu et al., in case of synchronous cancers, the most likely explanation seems to be a sporadic thyroid cancer on a background of genetic cancer predisposition made by Lynch syndrome [25,26]. In this setting, an increased predisposition to develop thyroid nodules could be present as well. Indeed, we found a higher prevalence of thyroid nodules than that seen in the general population.

In addition, the major neoplastic involvement in patients with thyroid nodules and previous history of thyroid disease compared to the group without thyroid nodules, may be a further sign of increased tumoral predisposition due to genetic weight.

We should also consider that TIR2 and TIR3a present a risk of malignancy, which in the general population is below 3% and 5–15%, respectively [10].

Based on these assumptions, although LS patients with simultaneous neck cancers are rare and thyroid nodules in Lynch syndrome seem to be primarily benign [27], screening of thyroid nodules by ultrasound should be evaluated, especially in MSH6 carrier patients, for which we found a higher prevalence of thyroid nodules.

Performing at least one thyroid ultrasound examination may be helpful for the detection of nodular thyroid disease in LS patients. FNA should be performed for all high-risk thyroid nodules and serial imaging is useful for the follow-up of suspicious US findings.

Furthermore, in cases of specific cancer predisposition due to familiar and/or environmental factors, there is a need to follow-up on LS patients for the development of rare associated cancers.

The limitations of this study are primarily associated with its retrospective design. In fact, it lacks longitudinal follow-up, which is crucial to understanding the progression and potential malignancy of the nodules. Secondly, the study did not perform genetic analysis on the neoplastic tissue of thyroid cancer cases, missing an opportunity to explore the molecular relationship between LS and thyroid cancer more deeply. Lastly, the research was conducted at a single center, which may limit the generalizability of the findings to a broader population.

## 5. Conclusions

A high prevalence of thyroid nodules is found in patients with LS, especially in MSH6 carrier patients. Performing at least one thyroid ultrasound examination is suggested for the detection of nodular thyroid disease in LS patients. Systematic investigations and multi-center studies with longitudinal follow-up are needed to estimate their prevalence, features, and risk of malignant transformation.

## Figures and Tables

**Figure 1 genes-15-00859-f001:**
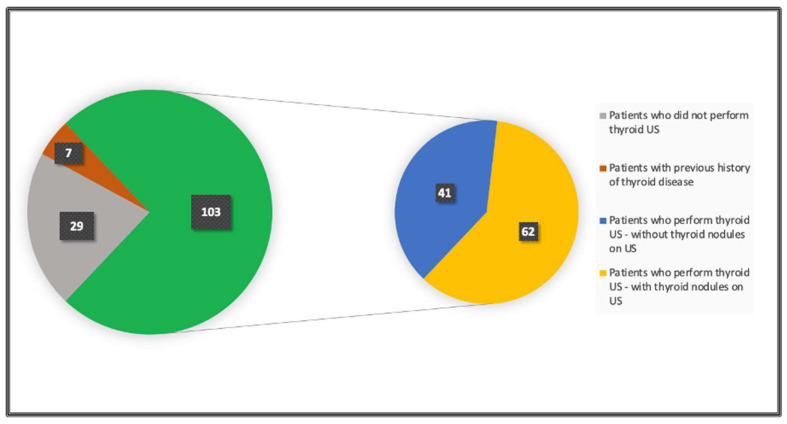
Groups of patients.

**Table 1 genes-15-00859-t001:** Demographic data. Values are expressed as *n* (%) or median (range).

	Total (*n* = 110)	Patients Who Presented Thyroid Nodules in US and with Previous History of Thyroid Disease (*n* = 69)	Patients without Thyroid Nodules in US (*n* = 41)
Female sex	75 (68)	53 (77)	22 (54)
Age, years	48.6 (12.3)	51.9 (11.6)	43 (11.6)
Pathogenic variant			
MSH6	37 (34)	25 (36)	12 (29)
MSH2	39 (35)	22 (32)	17 (41)
MLH1	28 (26)	17 (24)	11 (27)
PMS2	6 (5)	5 (7)	1 (3)
Affected branch of the family			
Maternal	61 (56)	36 (52)	25 (61)
Paternal	41 (37)	26 (38)	15 (37)
Not Available	8 (7)	7 (10)	1 (2)

**Table 2 genes-15-00859-t002:** Associated neoplastic diseases.

	Patients who Presented Thyroid Nodules on US and with Previous History of Thyroid Disease (*n*, %)	Patients without Thyroid Nodules on US (*n*, %)
Colorectal cancer	20 (29%)	6 (15%)
Endometrial cancer	22 (32%)	6 (15%)
Ovarian cancer	5 (7%)	1 (2.4%)
Synchronous endometrial and ovarian cancer	3 (4%)	2 (5%)
Skin cancer (basocellular carcinoma and melanoma)	6 (9%)	2 (5%)
Breast cancer	5 (7%)	3 (7%)
Urinary tract	2 (3%)	2 (5%)
Prostate cancer	2 (3%)	0 (0%)
Gastric cancer	1 (1.4%)	0 (0%)
Small bowel cancer	1 (1.4%)	1 (2.4%)

## Data Availability

The original contributions presented in the study are included in the article, further inquiries can be directed to the corresponding author.

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
