# Peer review of "Lynch Syndrome and Thyroid Nodules: A Single Center Experience"

_genes, 2024, doi:10.3390/genes15070859_

Round 1
Reviewer 1 Report
Comments and Suggestions for Authors
The authors, led by Irene Spinelli, conducted a study to investigate the presence of thyroid nodules in patients with Lynch Syndrome (LS) and their association with genetic features of the disease. They found a high prevalence of thyroid nodules, particularly in patients carrying the MSH6 mutation. The study suggests that systematic thyroid ultrasound examinations could be beneficial for LS patients to detect nodular thyroid disease early.
Major points
The study included a retrospective analysis of 139 LS patients, providing a substantial data set for examination.
A notable 60% of patients who underwent ultrasound had thyroid nodules, which is higher than the general population, indicating a potential link between LS and thyroid nodules.
The study not only reported the prevalence of nodules but also delved into the cytologic analysis, offering insights into the nature of these nodules.
Minor points
The research was conducted at a single center, which may limit the generalizability of the findings to a broader population.
As a retrospective study, it lacks longitudinal follow-up, which is crucial to understand the progression and potential malignancy of the nodules.
The study did not perform genetic analysis on the neoplastic tissue of thyroid cancer cases, missing an opportunity to explore the molecular relationship between LS and thyroid cancer more deeply.
These different points could be included as a limitation paragraph in the discussion.
Overall, the study provides valuable insights into the relationship between LS and thyroid nodules, suggesting a need for further research and consideration of thyroid screening in LS patients. However, the limitations highlight the necessity for more extensive, multi-center studies with longitudinal follow-up and genetic analysis of neoplastic tissues.
Comments on the Quality of English LanguageMinor typos.
Author Response
Limitations of this study are primarily associated with its retrospective design. In facts, it lacks longitudinal follow-up, which is crucial to understand the progression and potential malignancy of the nodules. Secondly, the study did not perform the genetic analysis on the neoplastic tissue of thyroid cancer cases, missing an opportunity to explore the molecular relationship between LS and thyroid cancer more deeply. Lastly, the research was conducted at a single center, which may limit the generalizability of the findings to a broader population.
5. Conclusions
A high prevalence of thyroid nodules is found in patients with LS, especially in MSH6 carrier patients. Performing at least one thyroid ultrasound examination is suggested for detection of nodular thyroid disease in LS patients. Systematic investigations and multi-center studies with longitudinal follow-up are needed to estimate their prevalence, features, and risk of malignant transformation.

Reviewer 2 Report
Comments and Suggestions for Authors
Spinelli et al performed a retrospective analysis to investigate the prevalence of thyroid nodules in Lynch Syndrome (LS) patients. The authors found that there is a high prevalence of thyroid nodules in LS patients (especially with MSH6 pathogenic variants). This finding suggests that a thyroid ultrasound examination may be useful in LS patients.
However, there are several issues that need to be addressed before publication:
1. Demographic information (except for age) is missing. For example, are thyroid nodules more common in males or females?
2. More information is also needed for the patients who had intermediate and high risk nodules and underwent FNA biopsy. What was their genetic profile?
3. Table 1: Are these differences statistically significant?
4. The limitations of the study need to be discussed.
5. It is not clear if all the patients with thyroid nodules discovered on US had single nodules or how many of them had multiple.
5. The manuscript needs significant editing for clarity and readability.
Comments on the Quality of English LanguageThe manuscript needs significant editing for clarity and readability.
Author Response
Spinelli et al performed a retrospective analysis to investigate the prevalence of thyroid nodules in Lynch Syndrome (LS) patients. The authors found that there is a high prevalence of thyroid nodules in LS patients (especially with MSH6 pathogenic variants). This finding suggests that a thyroid ultrasound examination may be useful in LS patients.
However, there are several issues that need to be addressed before publication:
1. Demographic information (except for age) is missing. For example, are thyroid nodules more common in males or females?
Reply: Thyroid nodules were found in 62 patients (60%) who underwent US, 46 were females and 16 were males. Of them, 22 (35%) had single nodule and 40 (65%) had multiple nodules.
2. More information is also needed for the patients who had intermediate and high risk nodules and underwent FNA biopsy. What was their genetic profile?
Reply: In particular, of the seven patients with intermediate and high risk nodules, 4 patients (57%) were carrier of MSH6, 2 (28%) of MLH1 and 1 (15%) of MSH2 mutation.
3. Table 1: Are these differences statistically significant?
Reply: There is no mention of differences. We just reported number of patients with associated neoplastic diseases
4. The limitations of the study need to be discussed.
Reply: Please see above
5. It is not clear if all the patients with thyroid nodules discovered on US had single nodules or how many of them had multiple.
Reply: Thyroid nodules were found in 62 patients (60%) who underwent US, 46 were females and 16 were males. Of them, 22 (35%) had single nodule and 40 (65%) had multiple nodules (> 1 nodule).

Round 2
Reviewer 2 Report
Comments and Suggestions for Authors
The authors addressed my comments in a satisfactory way.
Comments on the Quality of English LanguagePlease check the manuscript for typos.